# The Manufacturing Process of Lotus (*Nelumbo Nucifera*) Leaf Black Tea and Its Microbial Diversity Analysis

**DOI:** 10.3390/foods14030519

**Published:** 2025-02-06

**Authors:** Xiaojing Gao, Xuhui Kan, Fengfeng Du, Linhe Sun, Xixi Li, Jixiang Liu, Xiaojing Liu, Dongrui Yao

**Affiliations:** 1Jiangsu Key Laboratory for the Research and Utilization of Plant Resources, Institute of Botany, Jiangsu Province and Chinese Academy of Sciences, Nanjing Botanical Garden Mem. Sun Yat-Sen, Nanjing 210014, China; jing18905150107@126.com (X.G.); 2020208004@stu.njau.edu.cn (X.K.); duffyx@163.com (F.D.); linhesun@cnbg.net (L.S.); xixili0419@163.com (X.L.); ljx891654338@163.com (J.L.); 2Jiangsu Engineering Research Center for Landscape Plant Resources and Germplasm Innovation, Nanjing 210014, China

**Keywords:** lotus leaves, fermented black tea, polyphenol oxidase, microbial composition

## Abstract

Lotus leaves combine both edible and medicinal properties and are rich in nutrients and bioactive compounds. In this study, the lotus leaf tea was prepared using a black tea fermentation process, and the functional components and microbial changes during fermentation were investigated. The results indicated that the activity of polyphenol oxidase showed an initial rise followed by a decline as fermentation progressed, peaked at 3 h with 1.07 enzyme activity units during fermentation. The lotus leaf fermented tea has high levels of soluble sugars (20.92 ± 0.53 mg/g), total flavonoids (1.59 ± 0.05 mg GAE/g), and total polyphenols (41.34 ± 0.87 mg RE/g). Its antioxidant activity was evaluated using ABTS, DPPH, and hydroxyl radical scavenging assays, with results of 18.90 ± 1.02 mg Vc/g, 47.62 ± 0.51 mg Vc/g, and 17.58 ± 1.06 mg Vc/g, respectively. The microbial community also shifted during fermentation. *Fusarium* played a significant role during the fermentation process. This study demonstrated that the black tea fermentation process improved the functional components and biological activity of lotus leaf tea by optimizing the synergistic effect of enzymatic oxidation and microbial fermentation. The findings not only realized the comprehensive utilization of lotus leaf resources but also provided a foundation for developing innovative functional beverages with enhanced bioactive properties.

## 1. Introduction

Lotus (*Nelumbo nucifera* Gaertn.) is an important perennial aquatic plant with high edible, medicinal, and ornamental value [1]. Lotus leaves, a centuries-old component in traditional Chinese medicine, are not only known for their medicinal properties but also for their nutrient richness [2]. Lotus leaves are rich in bioactive compounds such as alkaloids, flavonoids, terpenoids, and polysaccharides [3,4], and they play an important role in antioxidant [5], lipid-reducing [6], anti-inflammatory [7], anti-cancer [8], and liver protection [9,10]. Therefore, lotus leaves are suitable for processing into health supplements or for inclusion in nutritional foods to leverage their dietary value [11]. In addition, the carbonization of lotus leaves can also be used to produce biochar with hemostatic effects in traditional Chinese medicine [12]. Although lotus leaves are abundant and inexpensive in China, they have not been fully utilized yet [13].

Tea is one of the most widely consumed beverages in the world. Based on the fermentation process, tea can be categorized into unfermented tea, slightly fermented tea, light fermented tea, semi-fermented tea, fully fermented tea, and post-fermented tea [14]. In addition to the traditional tea trees (*Camellia sinensis*), many other plants have also been developed into tea drinks with specific flavors and potential health benefits [15]. These plant teas, sometimes called herbal teas, may originate from the leaves, flowers, roots, fruits, or bark of plants [16]. Leaves of the lotus have been used as a tea ingredient in Asian countries, and lotus leaf tea is recognized as one of the most ancient teas, which are produced through methods similar to traditional green tea [17]. The processing methods for lotus leaf tea aim to preserve the natural compounds found in the leaves, such as flavonoids, alkaloids, and vitamins [18]. Unhealthy dietary habits, such as an excessive intake of sugar and fat, are likely to cause diseases such as obesity, diabetes, and non-alcoholic fatty liver disease. In addition, the incidence of neurological disorders is gradually increasing due to a number of environmental factors, such as stressful living conditions and poor sleep quality [19]. The global demand for functional teas is rapidly increasing, driven by health-conscious consumers seeking beverages with additional health benefits, such as improved digestion, stress relief, and detoxification. Lotus leaf holds a unique position in this market due to its abundance of bioactive compounds, including antioxidants, alkaloids, and dietary fiber, which support weight management, cardiovascular health, and relaxation [11]. Additionally, its sustainable cultivation and symbolic heritage enhance its appeal, making lotus leaf tea a premium choice in the growing functional tea industry.

Some health benefits of lotus leaf tea have been reported [11]. Researchers found that tender lotus leaves treated with the green tea process exhibited good antioxidant activity [20]. Lotus tea in Vietnam conformed to the international standards for tea and was found to have high biochemical diversity, which contributes to the positive impact on human health [21]. However, most of these studies were based on lotus leaf tea made using the green tea process, and few studies have been reported on lotus leaf fermented teas. Black tea refers to the fully fermented tea leaves. During the processing of black tea, enzymes in the tea cells catalyze the oxidation and polymerization of catechins to form the characteristic flavor substances [22,23,24]. Due to the processing method and resulting chemical transformations, fermented teas offer several advantages, such as mildness, digestive benefits, etc. [25]. Therefore, the fermentation technology may provide a new way to improve the nutritional and medicinal value of lotus leaves and help to develop and utilize the resources of lotus leaves. The key process in black tea production is the oxidation catalyzed by polyphenol oxidase, which leads to the reaction of polyphenols and other substances to form tea pigments and other key components, thereby altering the quality of the tea leaves [26]. As black tea fermentation proceeds, the microorganisms in the tea may affect the content of polyphenolic compounds as well as the antioxidant properties of the tea through their metabolic activities, thus altering the tea’s aroma, flavor, and health functions. Therefore, the changes in microorganisms during the fermentation process of black tea from lotus leaves are worthy of in-depth investigation.

The hypothesis of this study is that a controlled fermentation process can enhance the functional properties of lotus leaf tea by promoting the transformation of bioactive compounds. In this study, the fermentation tea technology was used to process lotus leaf, and the changes in key processing parameters and microbial community during the whole process were analyzed to determine the specific conditions for the fermentation of lotus leaf tea. Specifically, the processing technology was optimized by systematically monitoring the dynamics of key functional components and enzymatic activity throughout fermentation. Additionally, high-throughput sequencing was employed to analyze shifts in the microbial community, and antioxidant activity assays were conducted to assess improvements in bioactivity. This study aims to provide a theoretical basis for the processing technology of fermented lotus leaf tea and improve the value of lotus leaf resources.

## 2. Materials and Methods

### 2.1. Plant Materials

The selected lotus plants were cultivated in Institute of Botany, Jangsu Province and Chinese Academy of Science, Xuanwu District, Nanjing City, Jiangsu Province, China. Lotus leaves were used as experimental materials. Fresh lotus leaf samples at different stages were collected, including lotus leaves from the surface of the water to the beginning of the aging process.

### 2.2. Fermentation of the Lotus Leaf

The fermented tea-making process includes cutting, withering, rolling, fermentation, and drying. The lotus leaf samples were separated into small pieces, about 1 cm^2^. The lotus leaves were spread evenly in a pile about 2–3 cm thick and wilted at room temperature, about 25 °C, during which time the leaves were turned regularly. Leaves wilting at intervals of 0, 2, 3, 4, and 5 h were sampled. Then, the leaves were passed through a rolling machine (Changsha Oulong Machinery Equipment Trading Co., Ltd., Changsha, Hunan Province, China) for about an hour.

Then, the rolled lotus leaves were spread evenly on the well-ventilated fermentation frame (Kunshan Yongwang Plastic Products Co., Ltd., Suzhou, Jiangsu Province, China). The relative humidity was maintained at 90%, and the lotus leaves were fermented at a temperature of 35 °C. For sequencing, samples were taken during fermentation at intervals of 0, 3, 6, 12, and 24 h. Then, the samples were immediately frozen in liquid nitrogen (Nanjing Jindongsheng Trading Co., Ltd., Nanjing, Jiangsu Province, China) and stored at −80 °C for further analysis. For black tea making, fermented lotus leaves were further dried at 115 °C for 15 min to terminate fermentation via a tea dryer (Quanzhou Yike Machinery Co., Ltd., Quanzhou, Fujian Province, China). Then, the samples were further dried at 95 °C for about an hour. Photographs of lotus black tea at different stages was shown in Figure 1.

### 2.3. Determination of Key Enzyme Activity

The fresh lotus leaves samples were ground to disrupt the cellular structures, and the crude enzyme liquid was extracted following established methods [27]. The extraction solution contained 0.05 mol/L of pre-cooled citric acid–phosphate buffer (pH = 6.5) with a small amount of insoluble polyvinylpyrrolidone (PVP). A total of 0.5 g of lotus leaves were added to the pre-cooled mortar (Nanjing Kangluoda Experimental Technology Co., Ltd., Nanjing, Jiangsu Province, China); the extraction solution and quartz sand were then added, and it was ground into a homogenate and set to a volume of 7 mL. The extract was placed in the refrigerator (Midea, Foshan, Guangdong Province, China)and stirred several times at 4 °C for 12 h. The sample was centrifuged at 0 °C at 5000 rpm for 10 min, and the supernatant is the crude enzyme liquid.

Polyphenol oxidase (PPO) activity was determined using the colorimetric method with slight modifications [28]. The reaction mixture consisted of 0.6 mL of 1% catechol as substrate, 0.4 mL of 0.1% proline as stabilizers and accelerators, 2 mL of 0.1 mol/L citric acid–phosphate buffer (pH 5.6), and 1 mL of the enzyme solution. After incubation at 37 °C for 10 min, one unit of PPO enzyme activity was defined as a change in absorbance of 0.1 unit per minute in a 10 mm path at 420 nm.

### 2.4. Determination of Water Content

The water content of the samples was determined by drying using the direct measurement method. Briefly, about 5 g of lotus leaf samples were weighed in various drying dishes and placed in an oven dryer (Shaoxing Shangcheng Instrument Manufacturing Co., Ltd., Shaoxing, Zhejiang Province, China) at 120 °C, heated to achieve a constant weight and then cooled to room temperature. The water content of the lotus leaf sample was expressed as the percentage of mass difference before and after drying to total mass.

### 2.5. DNA Extraction and PCR Amplification

Microbial DNA was extracted from lotus leaves samples using the E.Z.N.A.^®^ Soil DNA Kit (Omega Bio-tek, Norcross, GA, USA) according to manufacturer’s protocols. The V4-V5 regions of the bacteria 16S ribosomal RNA gene were amplified using primers 515F 5′-barcode-GTGCCAGCMGCCGCGG)-3′ and 907R 5′-CCGTCAATTCMTTTRAGTTT-3′; the fungal ITS rRNA gene was amplified using the primer pair ITS1F (5′-CTTGGTCATTTAGAGGAAGTAA-3′) and ITS2 (5′-GCTGCGTTCTTCATCGATGC-3′). The PCR cycle consisted of initial denaturation at 95 °C for 2 min, followed by 25 cycles of denaturation at 95 °C for 30 s, annealing at 55 °C for 30 s, elongation at 72 °C for 30 s, and a final extension at 72 °C for 5 min, where the barcode is an eight-base sequence unique to each sample. PCR reactions were performed in triplicate. The 20 μL mixture contained 4 μL of 5×FastPfu Buffer, 2 μL of 2.5 mM dNTPs, 0.8 μL of each primer (5 μM), 0.4 μL of FastPfu Polymerase, and 10 ng of template DNA. Amplicons were extracted from 2% agarose gels and purified using the AxyPrep DNA Gel Extraction Kit (Axygen Biosciences, Union City, CA, USA).

Raw reads files were demultiplexed, quality filtered, and analyzed using the Quantitative Insights Into Microbial Ecology (QIIME, v1.7.0) pipeline. To minimize redundant calculations during analysis, non-repetitive sequences were selectively extracted from the sample’s optimal sequences. Operational taxonomic units (OTUs) were clustered with 97% similarity cutoff using the UPARSE software (version 7.1, http://drive5.com/uparse/, accessed on 4 December 2023), and chimeric sequences were identified and removed using UCHIME. The most abundant sequence from each OTU was selected to represent the OUT. OTUs were classified taxonomically using a QIIME-based wrapper from the Ribosomal Database Project (RDP) classifier against UNITE (User-friendly Nordic ITS Ectomycorrhiza Database) and INSD (International Nucleotide Sequence Databank) database [29].

### 2.6. Library Construction and Sequencing

Purified PCR products were quantified using Qubit^®^ 3.0 (Life Invitrogen) (Waltham, MA, USA), and every twenty-four amplicons whose barcodes were different were mixed equally. The pooled DNA product was used to construct an Illumina Pair-End library following Illumina’s genomic DNA library preparation procedure. Then, the amplicon library was paired-end sequenced (2 × 250) on an Illumina MiSeq platform (Shanghai BIOZERON Co., Ltd.) (Shanghai, China) according to the standard protocols. The resulting 16S and ITS amplicon reads from the Hi-Seq Illumina sequencing platform were imported.

### 2.7. Quality Chemical Composition Assay

The pH level of different states of lotus leaves was measured using a digital pH meter (Model pHS-25 PH Meter, Shanghai, China). The contents of soluble sugar, total polyphenols, and flavonoids were determined according to the reported methods [30,31,32]. The contents of total polyphenols and flavonoids were expressed as mg of gallic acid equivalent (GAE)/L and rutin equivalents (RE)/L, respectively.

The antioxidant capacity of different states of lotus leaves was comprehensively evaluated using three distinct methods: the 2,2′-Azino-bis(3-ethylbenzothiazoline-6-sulfonic acid) (ABTS) method, the 1,1-Diphenyl-2-picrylhydrazyl (DPPH) method, and the hydroxyl radical scavenging activity assay based on the Fenton reaction [33]. The ABTS method measures the ability of antioxidants to neutralize the ABTS radical cation, which is indicated by a decrease in absorbance at 734 nm. The DPPH method assesses the scavenging ability of antioxidants against the DPPH radical, which is determined by the reduction in absorbance at 517 nm. Lastly, the hydroxyl radical scavenging activity, determined by the Fenton reaction, provides insight into the capacity of tea broth to quench highly reactive hydroxyl radicals.

### 2.8. Statistical Analysis

The experiment was repeated three times, and the results of each experiment were expressed as the average of three repeats. Statistical analysis was carried out using a SPSS package (Version 25.0). One-way analysis of variance (ANOVA) with a probability level of 5% (*p* ≤ 0.05) indicated statistical significance among the different treatments, which depended on Duncan’s multiple-range test.

## 3. Results and Discussion

### 3.1. Selection of Lotus Leaves for Black Tea

In the production of black tea, the maturity of the leaves plays a pivotal role in determining the activity of PPO, which is essential for the tea fermentation process. Consequently, to facilitate the processing of lotus leaves, the samples were meticulously classified into defined maturity stages, ranging from the initial S1 stage to the fully mature S5 stage (Figure 2A). This categorization allows for the precise determination of PPO activity at each stage. The activity of PPO was significantly higher during the early stages of leaf development, particularly at S1 and S2, with an activity level of 1.09 and 1.26 units observed at the S1 and S2 stages. This level was more than double that observed at the S5 stage. The PPO activity experienced a sharp decline from S2 to S3, after which it stabilized at a comparatively low level of 0.56 units at the S4 and S5 stages (Figure 2B). PPO plays a crucial role in the processing of black tea, catalyzing the oxidation of catechins into theaflavins, thearubigins, and theabrownins, among others. The activity level of PPO directly influenced the quality of black tea, showing a decisive impact on its color, aroma, and taste, thus playing a key role in the sensory and quality attributes of the tea [34,35,36]. Therefore, selecting lotus leaves with high PPO activity for subsequent processing is essential. Previous researchers found that tender lotus leaves are in a rapid growth phase, with active cellular metabolism and high enzyme synthesis, resulting in relatively high PPO activity; the content of polyphenolic compounds in the tender leaves is also high [37]. Given the optimal conditions at the S2 stage for PPO activity, these leaves were selected as the ideal raw material for the subsequent crafting of lotus black tea.

### 3.2. Determination of Withering Time

The selection of the lotus leaf at the appropriate stage was followed by the withering operation. To investigate the impact of withering duration on the manufacturing process of lotus black tea, samples were collected at 0, 1, 2, 3, 4, and 5 h intervals. Accordingly, the pivotal enzyme PPO activity and the moisture levels of lotus leaves at each time point were measured.

During the withering process, the activity of PPO in lotus leaves followed a pattern of initial increase and subsequent decrease. At the critical 2 h and 3 h intervals, PPO activity underwent a pronounced increase, reaching peak levels of 1.5 to 1.6 enzyme activity units (Figure 3A). Simultaneously, the moisture content of lotus leaves exhibited a steady linear decrease. During the 2–3 h period of the highest PPO activity, the water content was maintained between 60% and 70%. This optimal moisture range is crucial as it renders the leaves more susceptible to the rolling process, an essential step that facilitates the subsequent fermentation stage in black tea production [38]. Appropriate water content can enhance the activity of enzymes in tea leaves, facilitating the transformation of internal components, such as the oxidation of tea polyphenols [39]. Excessive water content can lead to over-fermentation, which impacts the tea’s aroma and flavor; on the other hand, insufficient water can cause incomplete fermentation [40]. Consequently, a withering duration of 2 h was used for processing lotus leaf tea in this study.

### 3.3. Microbial Community Diversity During the Fermentation Process

The withered lotus leaves were then subjected to different degrees of fermentation. To elucidate the microbial composition throughout the fermentation process, 16S rRNA gene sequencing for bacterial communities and internal transcribed spacer (ITS) region sequencing for fungal communities were analyzed.

All optimized sequences were mapped to their respective OTU representative sequences, and those with greater than 97% similarity were selected to compile the OTU tables. The 16S amplicon data yielded 634,085 bacterial sequences, which were grouped into 113 OTUs. The fungal community sequences, numbering 603,970, were clustered into 2818 OTUs.

The alpha diversity of microbial community throughout the fermentation process was evaluated using the R software (vegan 2.0 package). Based on the Chao1 index, the fungal community showed a notable increase between the 12 h and 24 h marks of fermentation. The Simpson index indicated that there was a significant increase in bacterial diversity when lotus leaf was fermented for 12 and 24 h, compared to the control group with fermentation within 6 h (Figure 4).

### 3.4. Dynamic Changes in Microbial Species Composition

The fermentation process of lotus leaves was found to harbor diverse microbial communities. Specifically, the bacterial community was characterized by 12 phyla, 16 classes, 23 orders, 35 families, and 29 genera. The fungal community comprised 6 phyla, 29 classes, 79 orders, 169 families, and 351 genera throughout the fermentation process of lotus leaves.

The analysis of species identification was conducted across all groups to identify core OTUs, which were consistently detected in all replicate samples. The bacteria were predominantly composed of the phyla Proteobacteria (52%), Cyanophyta (40%), and Bacteroideta (8%) (Figure 5A). The Proteobacteria phylum exhibited an increasing trend during the fermentation process (Figure 5C). Some researchers found that prolonged fermentation can lead to a decrease in the pH value of the fermentation environment, which may provide a suitable environment for Proteobacteria [41]. In addition, nutrients in tea, such as carbon sources, can be converted into small-molecular-weight metabolites that are more readily accessible to microorganisms, which may promote an increase in Proteobacteria. The excessive proliferation of Proteobacteria caused by overly extended fermentation times may affect the quality of tea, especially its aroma and flavor [42]. Therefore, longer fermentation periods, such as 12 and 24 h, were not selected for future analysis. At the genus level of bacterial composition, the dominant genera during the withering and fermentation processes were from the order Cvanobacteriales, which accounting for 65.83%~80.08%, and the relative abundance decreased to 42.07% at 24 h. The proportion of *Pantoea* was present in initially very small amounts, ranging from 0.04% to 1.07%, but it experienced a sharp increase to a range of 7.77% to 20.79% after 12 h of fermentation. Subsequently, other low-abundance species, such as *Raoultella*, *Klebsiella*, and *Acinetobacter* were also detected at minimal levels early in the fermentation process, but their proportions increase to 9.17%, 3.03%, and 2.30%, respectively, after 24 h of fermentation. In the fungal core OTUs, the Ascomycetes (65.22%), Basidiomycetes (21.74%), and Mucor (13.04%) were the most prevalent phylum (Figure 5B). During the initial phase of fermentation, the proportion of the most abundant fungal Ascomycetes species steadily increased, reaching a peak within the first 6 h. Beyond this point, their proportion began to gradually decrease, continuing this trend throughout the remainder of the fermentation process (Figure 5D).

At the genus level of fungal composition, *Fusarium* emerged as the most dominant genus during the fermentation process. The relative abundance of *Fusarium* peaked at a maximum of 61.21% after 6 h of fermentation. However, it experienced a significant decline to 40.91% within 12 h and further dropped to 20.31% after 24 h. Although *Fusarium* is phytopathogenic for plants, the current publications indicated that the risk of mycotoxin contamination to health from tea consumption is negligible, except in some extreme cases [43]. Fungi were found mainly associated with non-volatile compounds in the fermented tea. The biotic stress brought about by *Fusarium* can promote the release or activity of relevant enzymes and thus facilitate the transformation of flavor substances. Studies have shown that *Fusarium* was found in high levels in oolong tea (Wuyi Rock Tea), and its presence may help to increase the theanine content of the tea, which in turn enhances the fresh flavor [44]. The genus of *Aspergillus* and *Penicillium* exhibited a significant rise, particularly at the 12 h and 24 h stages of fermentation. By the 24 h mark, *Aspergillus* reached its peak at 34.72%, while *Penicillium* also showed an increase, reaching 11.27% (Figure 5F). In addition, at the species level, *Fusarium* sp., which belongs to the genus *Fusarium*, accounted for a significant proportion, increasing from 59.30% at the stage of unfermented leaves to 72.57% at the 6 h fermentation stage, where it was the most dominant species. However, its prevalence began to decline at the 12 h stage and dropped to a relatively low level of 18.90% at the 24 h stage (Appendix A).

### 3.5. Co-Occurrence Network of Microbial Community

In the analysis of a microbial community sample, the co-occurrence network was established using Spearman correlation, which identified significant relationships between OTUs based on their relative abundances. To construct a co-occurrence network, Spearman correlation was applied to assess relationships between OTU pairs based on their relative abundances, with results subjected to FDR calibration to control for false discoveries. In constructing the co-occurrence network, the analysis was refined by initially filtering out OTUs with an average relative abundance below 0.01% or a detection rate below 60% to minimize false positives. The Spearman correlations with coefficients over 0.1 and FDR-adjusted *p*-values under 0.5 were retained, ensuring the inclusion of statistically significant microbial interactions.

In the co-occurrence networks of the fermentation process, the bacterial community exhibited a sparser structure with 11 nodes and 37 edges, resulting in an average degree of 6.73, indicating fewer connections per node (Appendix A). The fungal community displayed a more complex and interconnected network with 32 nodes and 452 edges, boasting an average degree of 28.25, reflecting a higher density of interactions among the nodes (Appendix A).

Key microbial species, like *Pantoea* and *Klebsiella* in bacteria and *Fusarium*, *Aspergillus*, and *Colletotrichum* in fungi, were identified as major nodes with high degrees and numerous edges, indicating their central roles. Additionally, *Bradyrhizobium* and *Enterobacter* in bacteria and *Heterodoassansia* and *Schizophyllum* in fungi were not the most predominant species, but they were still significant nodes with substantial connections. These results highlighted their essential function in shaping the microbial dynamics and interactions that propel the fermentation process. These microorganisms also played a significant role in influencing the metabolic products of lotus leaf tea. For instance, certain species of *Aspergillus* were widely used in the food industry for fermentation processes [45,46,47].

### 3.6. Microbial Beta Diversity Analysis

To analyze microbial beta diversity in the lotus leaf and the fermentation process, Principal Coordinates Analysis (PCoA) was utilized based on the Bray–Curtis distance of the samples, effectively visualizing the compositional disparities in microbial populations.

In the analysis of microbial beta diversity within the lotus leaf and the fermentation process, the distinct variations among bacterial communities across different groups were identified (Figure 6A). The analysis revealed that the original leaves group and the short fermentation time group (within 6 h) formed one distinct cluster, while the 12 h and 24 h fermentation groups formed other separate clusters. The first principal component (PC1) and the second principal component (PC2) explained 98% and 1% of the variability between the groups, respectively. These results indicated that the microbial communities of the original leaves and the early fermentation stage were closely related.

In the analysis of fungal communities across different groups, the PCoA revealed a dispersed distribution (Figure 6B). The original leaf group and the group before the fermentation begins clustered together, while the 3 h and 6 h short fermentation groups formed one cluster, and the 12 h and 24 h long fermentation groups formed another. This separation was significant, with the first principal coordinate (PC1) explaining 49% of the variance and the second (PC2) accounting for 15%. These results suggested that the fermentation of lotus leaves induces specific changes in the microbial community. Prolonged fermentation periods resulted in a further divergence in the microbial community composition, indicating that the duration of fermentation has a significant impact on the structure and abundance of the microorganisms present. This finding highlighted the importance of fermentation time in shaping the ecological dynamics and functional potential of the microbial communities associated with the lotus leaf fermentation process.

### 3.7. Functional Analysis Using PICRUSt 2

To predict the microbial community and determine the relative abundance of functional genes across different KEGG levels, PICRUSt (Phylogenetic Investigation of Communities by Reconstruction of Unobserved States) was employed, which utilizes the KEGG database. In this process, OTU representative sequences were integrated into the existing phylogenetic trees within the software for species-level annotation. The functional information was then derived using the IMG microbial genome database. The relative abundance of KEGG pathways across all samples or groups at levels 1, 2, and 3 was calculated based on the functional prediction results from PICRUSt2. Subsequently, statistical bar charts were generated for the KEGG pathway annotations at levels 1 and 2. This approach effectively identified the predominant metabolic pathways present in the study samples. Among the types of metabolic pathways identified, metabolism stands out as the most significant category. Notably, pathways associated with environmental information processing and genetic information processing also exhibited high relative abundances. Within the metabolic pathway category at level 2, carbohydrate metabolism and amino acid metabolism emerged as the predominant pathways. Furthermore, energy metabolism was observed to be relatively abundant among bacterial metabolic pathways (Figure 7).

### 3.8. Changes in the Activity of Key Enzymes During Processing

The results of PPO activity measurements during the processing and fermentation of lotus leaf tea are illustrated in Figure 8. After withering and rolling significantly increased PPO activity, the activity showed an initial rise followed by a decline as fermentation progressed. Throughout the entire processing process, the PPO activity was higher than that of lotus green tea (0.12 ± 0.01 U) (Appendix A). The maximum activity of 1.07 enzyme activity units was reached at the 3 h fermentation mark, which is approximately 1.6 times higher than at the onset of fermentation. At 6 h of fermentation, PPO activity was also significantly higher compared to other stages. However, by 12 h of fermentation, PPO activity had significantly decreased, and by 24 h, it had dropped to 0.42 enzyme activity units, lower than the initial enzyme activity at the commencement of fermentation. This indicated that PPO, a key enzyme, continues to play an important role at the 6 h fermentation mark, while extended fermentation leads to a decline in enzyme activity, suggesting that further fermentation may not be necessary. Combining the results of the microbial community analysis, which showed an increase in harmful microorganisms during extended fermentation, it can be concluded that a 6 h fermentation period is more appropriate. The trend of increasing and then decreasing PPO activity during withering and fermentation is primarily due to changes in substrate availability, environmental conditions, and enzymatic stability over time. At the beginning of withering and fermentation, cell disruption released phenolic substrates, which reacted with PPO, leading to increased enzyme activity [48]. However, as the process continues, these substrates were gradually depleted through oxidation or polymerization, causing a decline in PPO activity [49,50]. Additionally, PPO activity might be influenced by environmental factors such as temperature, humidity, and oxygen levels, which were initially favorable but may shift over time, reducing enzymatic efficiency [51,52].

### 3.9. Changes in Physical and Chemical Properties

Lotus leaves are rich in active components that significantly contribute to the flavor and quality of tea products [53]. The comparative composition of lotus leaf components across different fermentation stages is provided in Table 1. Following the original lotus leaves stages of withering and rolling, there was a notable increase in the soluble sugar and total flavonoids content, which are approximately 3 and 1.6 times higher than that of the original leaves, respectively. Concurrently, the total polyphenols content was maintained at a comparatively elevated level of 1.67 mg GAE/g.

At 6 h of fermentation, the content of various potent components in the tea was found to be higher. During the early stages of fermentation, polysaccharides were enzymatically broken down into simpler sugar molecules, resulting in an increase in soluble sugar content. However, as fermentation continues, these sugars were progressively metabolized by enzymes for microbial use, which can lead to a decrease in soluble sugar levels in the middle and later stages of the process [54]. Simultaneously, flavonoids are susceptible to oxidative polymerization, a process that leads to the formation of new compounds or macromolecules [55]. Consequently, this reaction resulted in a decrease in the overall flavonoid content. Similarly, tea polyphenols were subject to oxidation during fermentation, which led to a decrease in the overall polyphenol content. The pH value of lotus leaf tea initially increased in the early stages of fermentation, reaching a peak of 5.95 at 3 h, and then decreased as the fermentation period extends.

The initial increase in soluble sugars, total flavonoids, and polyphenols during fermentation may be due to mechanical processing, enzymes, and microbial activities, which released these compounds from the lotus leaves [56]. Cell wall degradation led to the release of sugars, flavonoids, and polyphenols. However, as fermentation progresses, these components may decline due to their consumption as substrates by microbes for energy and metabolism or their conversion into other non-detectable or less bioactive forms [40]. Additionally, prolonged oxidative reactions can lead to the polymerization or degradation of polyphenols and flavonoids, reducing their measurable levels.

The antioxidant activity results indicated that the ABTS radical scavenging capacity of the original leaves was significantly higher than that of other stages, reaching up to 23.70 mg Vc/g, and remained relatively high during the early stages of fermentation, dropping to 7.36 and 6.68 mg Vc/g at 12 and 24 h of fermentation, respectively. Conversely, the DPPH radical scavenging ability of the lotus leaves showed a clear increasing trend with extended processing and fermentation times, stabilizing at a high level of 47.35–48.44 mg Vc/g after 6 h of fermentation. The hydroxyl radical scavenging activity decreased to its lowest point after the withering and rolling process; then, it showed a clear trend of increasing and then decreasing with the start of fermentation, reaching a maximum of 20.68 mg Vc/g at the 12 h mark. The reducing power of the fermented lotus leaf tea between 6 and 24 h was significantly higher than that of the original leaves (Table 2). The antioxidant capacity of tea is closely related to its main components. The decline in antioxidant activity after 6 h of fermentation can be attributed to the degradation or transformation of bioactive compounds such as polyphenols and flavonoids, which initially contribute to antioxidant activity but may undergo oxidation, polymerization, or microbial metabolism over time [57]. Additionally, shifts in the microbial community may result in certain microbes utilizing these compounds as nutrients, further diminishing their levels. Notably, the total flavonoid content decreased significantly at 12 and 24 h of fermentation, leading to a decline in the antioxidant capacity. For example, the ABTS radical scavenging capacity drops from 18.90 ± 1.02 mg Vc/g at 6 h to 7.36 ± 1.49 mg Vc/g at 12 h and 6.68 ± 1.70 mg Vc/g at 24 h. These results indicated that the fermentation process in tea processing can enhance the overall antioxidant capacity of lotus leaf tea to some extent [58]. Additionally, it was found that the antioxidant capacity of fermented lotus leaf tea is higher than that of lotus leaf green tea (ABTS, DPPH, and hydroxyl radical scavenging activity were 8.07 ± 1.30 mg Vc/g, 45.01 ± 0.39 mg Vc/g, and 10.58 ± 3.31 mg Vc/g, respectively) (Appendix A), indicating an innovative advantage of using black tea fermentation processes for lotus leaf tea.

## 4. Conclusions

This study innovatively applied black tea fermentation technology to lotus leaf processing. Results indicated that using tender, unexpanded young lotus leaves and a process of withering for 2 h followed by fermentation for 6 h yielded the best quality fermented lotus leaf tea. The fermentation process significantly improved the functional components, such as polyphenols (1.59 ± 0.05 mg GAE/g) and flavonoids (41.34 ± 0.87 mg RE/g), and antioxidant activity of lotus leaf tea by optimizing fermentation conditions. In addition, the process of microbial communities was comprehensively analyzed through high-throughput sequencing. *Fusarium* was the dominant genus in the microbial community, reaching a peak relative abundance of 61.21% after 6 h. Metabolic pathways, particularly those involved in carbohydrate and amino acid metabolism, were significantly influenced by fermentation. The results provided a new method for the comprehensive utilization of lotus leaves and a practical guidance for the development of functional fermented tea drinks. However, the health-promoting effects of lotus leaf black tea were not investigated in depth in this study, and an assessment of taste and sensory aspects is lacking. In the future, further research will be conducted on the effects of lotus leaf components on the aroma, taste, and functionality of black tea to create sensory characteristics that are different from those of traditional black tea and to satisfy consumer demand for functionality and variety in tea products.

## Figures and Tables

**Figure 1 foods-14-00519-f001:**

Photographs of lotus black tea at different stages.

**Figure 2 foods-14-00519-f002:**
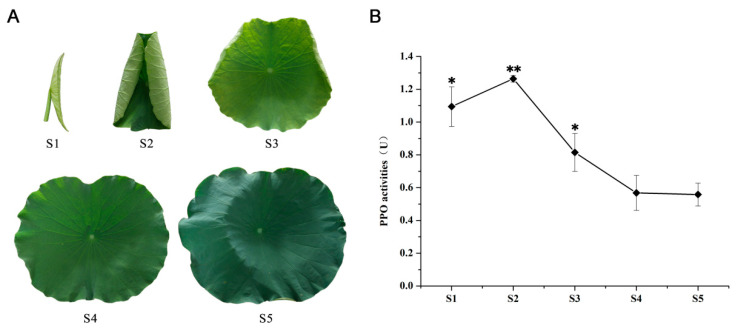
PPO activities at different developmental stages of lotus leaves. (**A**) Leaf developmental stages from S1 to S5. (**B**) Chart of PPO activities. The asterisk indicates a significant difference (single asterisk, *p* < 0.05; double asterisks, *p* < 0.01).

**Figure 3 foods-14-00519-f003:**
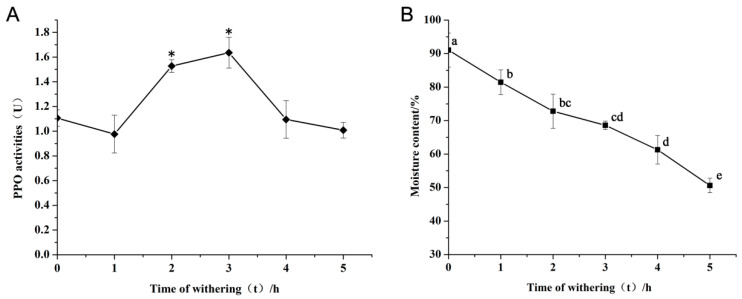
Parameters of withering leave. (**A**) PPO activities at different withering times for lotus leaves. (**B**) Moisture content at different withering times of lotus leaves. The asterisk indicates a significant difference (single asterisk, *p* < 0.05; different small letters in the same column indicate a significant difference at *p* < 0.05 level).

**Figure 4 foods-14-00519-f004:**
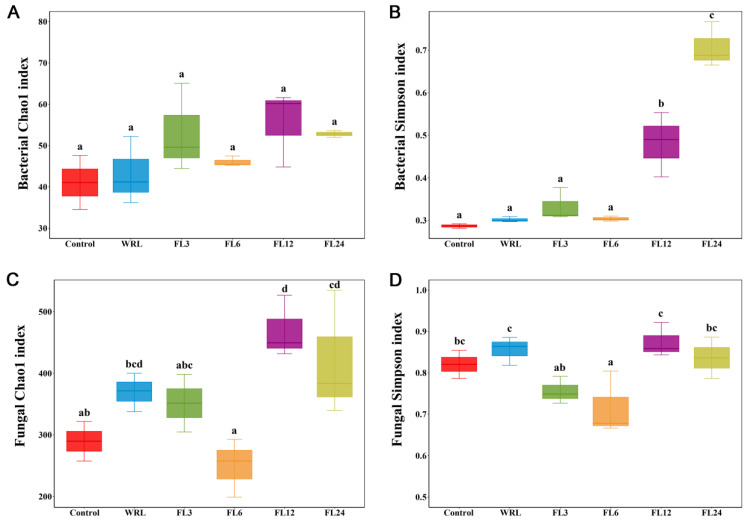
Microbial alpha diversity. Changes in the Chao1 and Simpson indexes of bacterial (**A**,**B**) and fungal (**C**,**D**) communities during the fermentation process of lotus leaves (different small letters in the same column indicate a significant difference at *p* < 0.05 level). WRL, withering and rolled leaves; FL3, leaves fermented for 3 h; FL6, leaves fermented for 6 h; FL12, leaves fermented for 12 h; FL24, leaves fermented for 24 h.

**Figure 5 foods-14-00519-f005:**
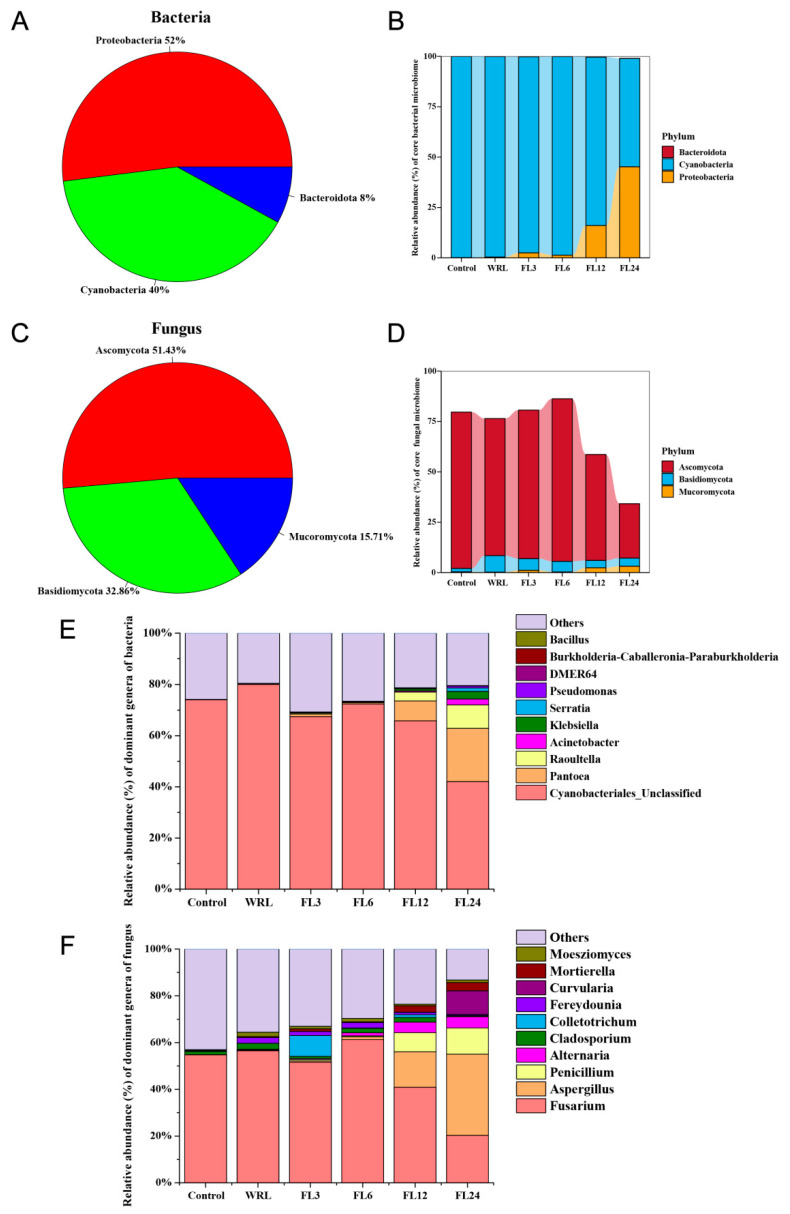
Microbial species composition. Community structure components of dominant species during the fermentation process of lotus leaves. Percentage and relative abundance (%) of the phyla in the core microbiome of bacteria (**A**,**B**) and fungus (**C**,**D**). Microbial taxonomic compositions showing the microbial successions at the genus level: (**E**) bacteria and (**F**) fungus. The results showed the top10 species, and the other species were classified into “Others”. WRL, withering and rolled leaves; FL3, leaves fermented for 3 h; FL6, leaves fermented for 6 h; FL12, leaves fermented for 12 h; FL24, leaves fermented for 24 h.

**Figure 6 foods-14-00519-f006:**
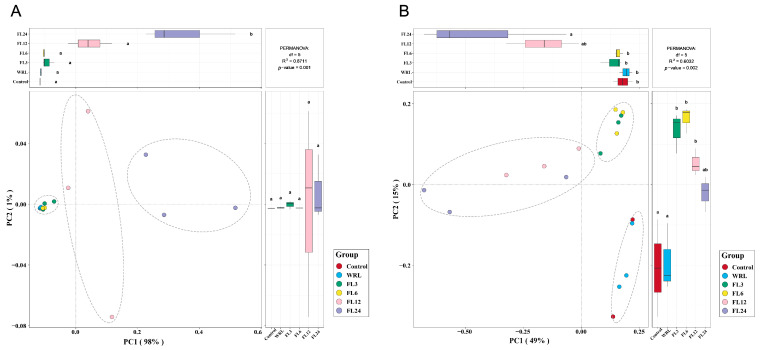
PCoA analysis with tests of microbial communities in samples during the fermentation process of lotus leaves: bacterial (**A**) and fungal (**B**). Based on the Bray-Curtis distance of the samples, the horizontal axis represents one principal component, and the vertical axis represents another principal component. The percentages indicate the contribution of the principal components to the differences among the samples. The samples are classified using dashed circles, and vertical/horizontal dashed lines represent the zero axes (different small letters in the same column indicate a significant difference at *p* < 0.05 level). WRL, withering and rolled leaves; FL3, leaves fermented for 3 h; FL6, leaves fermented for 6 h; FL12, leaves fermented for 12 h; FL24, leaves fermented for 24 h.

**Figure 7 foods-14-00519-f007:**
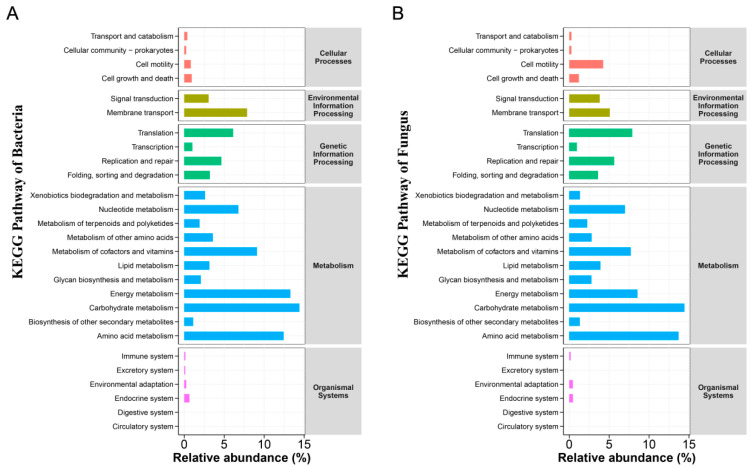
Prediction of the microbial function during the fermentation process of lotus leaves using PICRUSt (Phylogenetic Investigation of Communities by Reconstruction of Unobserved States): bacterial (**A**) and fungal (**B**). Average relative abundance of Kyoto Encyclopedia of Genes and Genomes (KEGG) pathway annotations at levels 1 and 2 for all lotus leaves samples. The colors of the bars represent different classification levels of Level 1, with annotations provided on the right side. The length of the bars represents the average relative abundance of the Level 2.

**Figure 8 foods-14-00519-f008:**
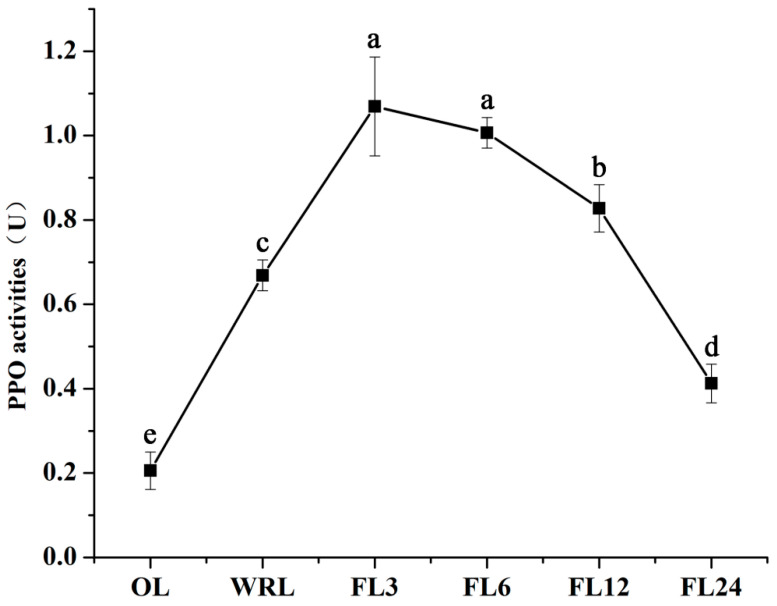
PPO activities during processing (different small letters in the same column indicate a significant difference at *p* < 0.05 level). OL, original leaves; WRL, withering and rolled leaves; FL3, leaves fermented for 3 h; FL6, leaves fermented for 6 h; FL12, leaves fermented for 12 h; FL24, leaves fermented for 24 h.

**Table 1 foods-14-00519-t001:** Contents of soluble sugar, total polyphenols, and total flavonoids and the power of hydrogen (pH) at different states in lotus leaves.

Group	Soluble Sugar (mg/g)	Total Polyphenols (mg GAE/g)	Total Flavonoids (mg RE/g)	pH
OL	11.56 ± 0.87 ^c^	1.67 ± 0.05 ^a^	25.67 ± 0.43 ^b^	5.76 ± 0.04 ^c^
WRL	34.14 ± 3.54 ^a^	1.67 ± 0.19 ^a^	41.65 ± 1.28 ^a^	5.82 ± 0.04 ^b^
FL3	19.42 ± 2.18 ^b^	1.17 ± 0.11 ^b^	25.85 ± 0.35 ^b^	5.95 ± 0.03 ^a^
FL6	20.92 ± 0.53 ^b^	1.59 ± 0.05 ^a^	41.34 ± 0.87 ^a^	5.61 ± 0.02 ^d^
FL12	7.77 ± 1.20 ^c^	1.23 ± 0.04 ^b^	26.36 ± 0.60 ^b^	5.30 ± 0.02 ^f^
FL24	7.86 ± 0.81 ^c^	1.01 ± 0.02 ^b^	18.98 ± 1.78 ^c^	5.37 ± 0.02 ^e^

Values were expressed as the mean ± standard deviation (SD). Different small letters in the same column indicated a significant difference at *p* < 0.05 level. OL, original leaves; WRL, withering and rolled leaves; FL3, leaves fermented for 3 h; FL6, leaves fermented for 6 h; FL12, leaves fermented for 12 h; FL24, leaves fermented for 24 h. GAE, gallic acid equivalent; RE, rutin equivalents.

**Table 2 foods-14-00519-t002:** Antioxidant activity at different states in lotus leaves.

Group	ABTS (mg Vc/g)	DPPH (mg Vc/g)	Hydroxyl Radical Scavenging Activity (mg Vc/g)
OL	23.70 ± 4.49 ^a^	42.68 ± 1.55 ^c^	13.59 ± 1.20 ^c^
WRL	15.33 ± 1.38 ^b^	45.62 ± 0.39 ^b^	10.72 ± 0.60 ^d^
FL3	17.82 ± 1.62 ^b^	45.88 ± 0.28 ^b^	13.44 ± 0.92 ^c^
FL6	18.90 ± 1.02 ^b^	47.62 ± 0.51 ^a^	17.58 ± 1.06 ^b^
FL12	7.36 ± 1.49 ^c^	48.44 ± 0.46 ^a^	20.68 ± 0.96 ^a^
FL24	6.68 ± 1.70 ^c^	47.35 ± 0.26 ^a^	18.02 ± 0.92 ^b^

Values were expressed as the mean ± standard deviation (SD). Different small letters in the same column indicated a significant difference at *p* < 0.05 level. OL, original leaves; WRL, withering and rolled leaves; FL3, leaves fermented for 3 h; FL6, leaves fermented for 6 h; FL12, leaves fermented for 12 h; FL24, leaves fermented for 24 h. Vc, Vitamin C.

## Data Availability

The original contributions presented in the study are included in the article/Appendix A; further inquiries can be directed to the corresponding authors. Data will be made available on request.

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
