# Peer review of "The Manufacturing Process of Lotus (Nelumbo Nucifera) Leaf Black Tea and Its Microbial Diversity Analysis"

_foods, 2025, doi:10.3390/foods14030519_

Round 1

Reviewer 1 Report

Comments and Suggestions for Authors

The study show novelty regarding the fermentation of lotus tea and provide insight into biochemical, microbial, and functional analyses, offering valuable insights into the process and outcomes.

Figures and tables are well-organized and effectively present the findings, particularly the enzyme activities and microbial diversity.

Introduction: consider to add more context regarding the global demand for functional teas and the uniqueness of using lotus leaves. This can help justify the study's relevance.

Discussion: consider to add the role of Fusarium during tea fermentation and their role in safety and the quality of the tea.

regarding the PPO activity and antioxidant activity, consider to compare with other types of tea (black tea or green tea).

In the conclusion, consider to acknowledge the gap of this research and suggest for future study.

while the data was well written, please ensure the manuscript follows the journal guidelines

Author Response

Response to Reviewer 1 Comments

1. Summary

Thank you very much for taking the time to review this manuscript. Please find the detailed responses below and the corresponding revisions in track changes in the re-submitted files.

2. Questions for General Evaluation

Reviewer’s Evaluation

Response and Revisions

Does the introduction provide sufficient background and include all relevant references?

Yes

Thank you for your comments.

Is the research design appropriate?

Yes

Thank you for your comments.

Are the methods adequately described?

Yes

Thank you for your comments.

Are the results clearly presented?

Can be improved

Thank you for your comments. We have improved the results in the revised manuscript.

Are the conclusions supported by the results?

Can be improved

Thank you for your comments. We have improved the conclusions in the revised manuscript.

3. Point-by-point response to Comments and Suggestions for Authors

Comments 1: The study show novelty regarding the fermentation of lotus tea and provide insight into biochemical, microbial, and functional analyses, offering valuable insights into the process and outcomes.

Response 1: We sincerely thank you for your careful review and constructive suggestions. Your interest in our work really gives us more confidence and motivates us to do better in the future.

Comments 2: Figures and tables are well-organized and effectively present the findings, particularly the enzyme activities and microbial diversity.

Response 2: We sincerely thank you for your careful review and encouragement.

Comments 3:Introduction: consider to add more context regarding the global demand for functional teas and the uniqueness of using lotus leaves. This can help justify the study's relevance.

Response 3:We sincerely thank you for your careful review and valuable advice. We have provided the information of the global demand for functional teas and the uniqueness of using lotus leaves in the revised Introduction (Line 50-61).

Unhealthy dietary habits, such as excessive intake of sugar and fat, are likely to cause diseases such as obesity, diabetes and non-alcoholic fatty liver disease. In addition, the incidence of neurological disorders is gradually increasing due to a number of environmental factors, such as stressful living conditions and poor sleep quality. The global demand for functional teas is rapidly increasing, driven by health-conscious consumers seeking beverages with additional health benefits, such as improved digestion, stress relief, and detoxification. Lotus leaf holds a unique position in this market due to its abundance of bioactive compounds, including antioxidants, alkaloids, and dietary fiber, which support weight management, cardiovascular health, and relaxation. Additionally, the sustainable cultivation and symbolic heritage enhance its appeal, making lotus leaf tea a premium choice in the growing functional tea industry.

Comments 4:Discussion: consider to add the role of Fusarium during tea fermentation and their role in safety and the quality of the tea.

Response 4:Thank you for your valuable advice. We have provided relevant information of Fusarium in the revised Discussion according to your advice (Line 287-295).

Although Fusarium is phytopathogenic for plant, the current publications indicated that the risk of mycotoxin contamination to health from tea consumption is negligible, except in some extreme cases. Fungi were found mainly associated with non-volatile compounds in the fermented tea. The biotic stress brought about by Fusarium can promote the release or activity of relevant enzymes, and thus facilitate the transformation of flavor substances. Studies have shown that Fusarium was found in high levels in oolong tea (Wuyi Rock Tea), and its presence may help to increase the theanine content of the tea, which in turn enhances the fresh flavor.

Comments 5:regarding the PPO activity and antioxidant activity, consider to compare with other types of tea (black tea or green tea).

Response 5:Thank you for your nice advice. We have compared the PPO activity and antioxidant activity of lotus tea with green tea (Line 379-380, 453-457). Revelant materials were provided in the Supplementary Materials (Table S1).

Throughout the entire processing process, the PPO activity was higher than that in lotus green tea (0.12±0.01 U) (Table S1).

Additionally, it was found that the antioxidant capacity of fermented lotus leaf tea is higher than that of lotus leaf green tea (ABTS, DPPH, hydroxyl radical scavenging activity were 8.07±1.30 mg Vc/g, 45.01±0.39 mg Vc/g, 10.58±3.31 mg Vc/g respectively) (Table S1), indicating an innovative advantage of using black tea fermentation processes for lotus leaf tea.

Comments 6:In the conclusion, consider to acknowledge the gap of this research and suggest for future study.

Response 6:Thank you for your nice advice. We have acknowledged the gap of this research and provided suggestion for future study (Line 471-476).

However, the health-promoting effects of lotus leaf black tea were not investigated in depth in this study and an assessment of taste and sensory aspects is lacked. In the future, further research will be conducted on the effects of lotus leaf components on the aroma, taste and functionality of black tea to create sensory characteristics that are different from those of traditional black tea, and to satisfy consumer demand for functionality and variety in tea products.

Comments 7:while the data was well written, please ensure the manuscript follows the journal guidelines

Response 7:Thanks for your kind reminder. We have checked and revised the manuscript to make sure it follows the journal guidelines.

4. Response to Comments on the Quality of English Language

Point 1:

Response 1:

5. Additional clarifications

No.

Reviewer 2 Report

Comments and Suggestions for Authors

In present study, the lotus leaf tea was prepared using a black tea fermentation process, and the functional components and microbial changes during fermentation were investigated. The results indicated that the changes during the fermentation process and the lotus leaf tea contained high levels of soluble sugars, total flavonoids, and total polyphenols, as well as good antioxidant activity. The microbial community also shifted during fermentation. Fusarium played a significant role during the fermentation process. The activity of polyphenol oxidase was measured during the processing, and the microbial composition was analyzed. The manuscript is interesting, but it needs major revision according to the following remarks.

The abstract should be improved. The novelty of the fermentation method should be reported and the major findings should be mentioned with obtained values, not just with noted trends of changes during the fermentation process.

The references should be organized in accordance with the journal specifications and there are some other issues as well (e.g. Latin names, the position of figures in the manuscript, others).

At the end of introduction part there is need to clearly state the novelty of performed research. What was done for the first time? Namely, it is written that the fermentation tea technology was used to process lotus leaf, and the changes of key processing parameters and microbial community during the whole process were analyzed to determine the specific conditions of the fermentation lotus leaf tea. However, it is not clear what is contribution of the research to already known data. In addition, the research hypothesis is missing and how is it to be tested. The novelty should be pointed out through the discussion part as well in the conclusions.

Lines 97-98: The sentence about extraction is not clear and it should be revised. In addition, the description of the experimental work should not be described as “Take about…”. The passive format should be used.

 The authors should discuss possible reasons for the fermentation process that initially increased the activity and levels of soluble sugars, total flavonoids, and polyphenols, with a subsequent decline, peaking in antioxidant activity after 6 h. What could be the reason for decline of antioxidant activity after 6h? 

Why PPO enzyme activity showed a trend of increasing and then decreasing during both withering and fermentation? The authors should provide possible explanations.

Author Response

Response to Reviewer 2 Comments

1. Summary

Thank you very much for taking the time to review this manuscript. Please find the detailed responses below and the corresponding revisionsin track changes in the re-submitted files.

2. Questions for General Evaluation

Reviewer’s Evaluation

Response and Revisions

Does the introduction provide sufficient background and include all relevant references?

Yes

Thank you for your comments.

Is the research design appropriate?

Can be improved

Thank you for your comments. We have improved the research design in the revised manuscript.

Are the methods adequately described?

Must be improved

We really thank you for your comments. We have improved the methods in the revised manuscript.

Are the results clearly presented?

Can be improved

Thank you for your comments. We have improved the results in the revised manuscript.

Are the conclusions supported by the results?

Yes

Thank you for your comments.

3. Point-by-point response to Comments and Suggestions for Authors

Comments 1: In present study, the lotus leaf tea was prepared using a black tea fermentation process, and the functional components and microbial changes during fermentation were investigated. The results indicated that the changes during the fermentation process and the lotus leaf tea contained high levels of soluble sugars, total flavonoids, and total polyphenols, as well as good antioxidant activity. The microbial community also shifted during fermentation. Fusarium played a significant role during the fermentation process. The activity of polyphenol oxidase was measured during the processing, and the microbial composition was analyzed. The manuscript is interesting, but it needs major revision according to the following remarks.

Response 1: We sincerely thank you for your careful review and constructive suggestions. Your interest in our work really gives us more confidence and motivates us to do better in the future. According to your nice advice, we have revised and checked carefully the entire manuscript.

Comments 2: The abstract should be improved. The novelty of the fermentation method should be reported and the major findings should be mentioned with obtained values, not just with noted trends of changes during the fermentation process.

Response 2: Thank you for your nice advice. We have added the novelty and the major findings according to your advice (Line 11-25).

Lotus leaves combine both edible and medicinal properties and are rich in nutrients and bioactive compounds. In this study, the lotus leaf tea was prepared using a black tea fermentation process, and the functional components and microbial changes during fermentation were investigated. The results indicated that the activity of polyphenol oxidase showed an initial rise followed by a decline as fermentation progressed, peaked at 3 h of 1.07 enzyme activity units during fermentation. The lotus leaf fermented tea has high levels of soluble sugars (20.92±0.53 mg/g), total flavonoids (1.59±0.05 mg GAE/g), and total polyphenols (41.34±0.87 mg RE/g). The antioxidant activity was evaluated using ABTS, DPPH, and hydroxyl radical scavenging assays, with results of 18.90±1.02 mg Vc/g, 47.62±0.51 mg Vc/g, and 17.58±1.06 mg Vc/g, respectively. The microbial community also shifted during fermentation. Fusarium played a significant role during the fermentation process. This study demonstrated that the black tea fermentation process improved the functional components and biological activity of lotus leaf tea by optimizing the synergistic effect of enzymatic oxidation and microbial fermentation. The findings not only realized the comprehensive utilization of lotus leaf resources but also provided a foundation for developing innovative functional beverages with enhanced bioactive properties.

Comments3: The references should be organized in accordance with the journal specifications and there are some other issues as well (e.g. Latin names, the position of figures in the manuscript, others).

Response 3: Thank you for your careful review. We have checked the references carefully to make sure they are in accordance with the journal specifications.

Comments 4: At the end of introduction part there is need to clearly state the novelty of performed research. What was done for the first time? Namely, it is written that the fermentation tea technology was used to process lotus leaf, and the changes of key processing parameters and microbial community during the whole process were analyzed to determine the specific conditions of the fermentation lotus leaf tea. However, it is not clear what is contribution of the research to already known data. In addition, the research hypothesis is missing and how is it to be tested. The novelty should be pointed out through the discussion part as well in the conclusions.

Response 4: Thank you for your careful review and nice advice. We have pointed out the novelty in the revised Introduction (Line 82-91), Discussion (Line 453-457) and Conclusions (Line 459-471). Also, the research hypothesis was added (Line 82-84) according to your advice.

Introduction: The hypothesis of this study is that a controlled fermentation process can enhance the functional properties of lotus leaf tea by promoting the transformation of bioactive compounds. In this study, the fermentation tea technology was used to process lotus leaf, and the changes of key processing parameters and microbial community during the whole process were analyzed to determine the specific conditions of the fermentation lotus leaf tea. Specifically, the processing technology was optimized by systematically monitoring the dynamics of key functional components and enzymatic activity throughout fermentation. Additionally, high-throughput sequencing was employed to analyze shifts in the microbial community, and antioxidant activity assays were conducted to assess improvements in bioactivity.

Discussion: Additionally, it was found that the antioxidant capacity of fermented lotus leaf tea is higher than that of lotus leaf green tea (ABTS, DPPH, hydroxyl radical scavenging activity were 8.07±1.30 mg Vc/g, 45.01±0.39 mg Vc/g, 10.58±3.31 mg Vc/g respectively) (Table S1), indicating an innovative advantage of using black tea fermentation processes for lotus leaf tea.

Conclusion: This study innovatively applied black tea fermentation technology to lotus leaf processing. Results indicated that using tender, unexpanded young lotus leaves and a process of withering for 2 h followed by fermentation for 6 h yielded the best quality fermented lotus leaf tea. The fermentation process significantly improved the functional components, such as polyphenols (1.59±0.05 mg GAE/g) and flavonoids (41.34±0.87 mg RE/g), and antioxidant activity of lotus leaf tea by optimizing fermentation conditions. In addition, the process of microbial communities was comprehensively analyzed through high-throughput sequencing. Fusarium was the dominant genus in the microbial community, reaching a peak relative abundance of 61.21% after 6 h. Metabolic pathways, particularly those involved in Carbohydrate and Amino Acid Metabolism, were significantly influenced by fermentation. The results provided a new method for the comprehensive utilization of lotus leaves and a practical guidance for the development of functional fermented tea drinks.

Comments 5: Lines 97-98: The sentence about extraction is not clear and it should be revised. In addition, the description of the experimental work should not be described as “Take about…”. The passive format should be used.

Response 5: Sorry for our carelessness. We have revised the extraction part to make it clearer. Also, the passive format was used in the revised description (Line 116-123).

The fresh lotus leaves samples were ground to disrupt cellular structures, and the crude enzyme liquid were extracted following established methods. The extraction solution contained 0.05 mol/L of pre-cooled citric acid-phosphate buffer (pH=6.5) with a small amount of insoluble polyvinylpyrrolidone (PVP). 0.5 g of lotus leaves were taken in the pre-cooled mortar, extraction solution and quartz sand were added and ground into a homogenate and set the volume to 7 mL. The extract was placed in the refrigerator and stir several times at 4℃ for 12 h. Centrifuge the sample at 0℃ 5000 rpm for 10 min, and the supernatant is the crude enzyme liquid.

Comments 6: The authors should discuss possible reasons for the fermentation process that initially increased the activity and levels of soluble sugars, total flavonoids, and polyphenols, with a subsequent decline, peaking in antioxidant activity after 6 h. What could be the reason for decline of antioxidant activity after 6h?

Response 6: Thanks for your careful review and nice advice. We have discussed the possible reasons for the change of antioxidant activity, levels of soluble sugars, levels of total flavonoids, and levels of polyphenols during the fermentation process in the revised manuscript (Line 422-430, 442-451).

The initial increase in soluble sugars, total flavonoids, and polyphenols during fermentation during fermentation may be due to mechanical processing, enzymes and microbial activities, which released these compounds from the lotus leaves. Cell wall degradation led to the release of sugars, flavonoids, and polyphenols. However, as fermentation progresses, these components may decline due to their consumption as substrates by microbes for energy and metabolism or their conversion into other non-detectable or less bioactive forms. Additionally, prolonged oxidative reactions can lead to the polymerization or degradation of polyphenols and flavonoids, reducing their measurable levels.

The antioxidant capacity of tea is closely related to its main components. The decline in antioxidant activity after 6 h of fermentation can be attributed to the degradation or transformation of bioactive compounds such as polyphenols and flavonoids, which initially contribute to antioxidant activity but may undergo oxidation, polymerization, or microbial metabolism over time. Additionally, shifts in the microbial community may result in certain microbes utilizing these compounds as nutrients, further diminishing their levels. Notably, the total flavonoid content decreased significantly at 12 and 24 h of fermentation, leading to a decline in the antioxidant capacity. For example, the ABTS radical scavenging capacity drops from 18.90±1.02 mg Vc/g at 6 h to 7.36±1.49 mg Vc/g at 12 h and 6.68±1.70 mg Vc/g at 24 h.

Comments 7: Why PPO enzyme activity showed a trend of increasing and then decreasing during both withering and fermentation? The authors should provide possible explanations.

Response 7: Thanks for your careful review and nice advice. We have provided possible explanations in the revised manuscript according to your advice (Line 391-400).

The trend of increasing and then decreasing PPO activity during withering and fermentation is primarily due to changes in substrate availability, environmental conditions, and enzymatic stability over time. At the beginning of withering and fermentation, cell disruption released phenolic substrates, which reacted with PPO, leading to increased enzyme activity. However, as the process continues, these substrates were gradually depleted through oxidation or polymerization, causing a decline in PPO activity. Additionally, PPO activity might be influenced by environmental factors such as temperature, humidity, and oxygen levels, which were initially favorable but may shift over time, reducing enzymatic efficiency.

4. Response to Comments on the Quality of English Language

Point 1:

Response 1:

5. Additional clarifications

No.

Round 2

Reviewer 1 Report

Comments and Suggestions for Authors

the authors have improved their content

Reviewer 2 Report

Comments and Suggestions for Authors

According to the changes made in the manuscript it can be accepted for publication.